# The Mechanisms and Efficacy of Photobiomodulation Therapy for Arthritis: A Comprehensive Review

**DOI:** 10.3390/ijms241814293

**Published:** 2023-09-19

**Authors:** Renlong Zhang, Junle Qu

**Affiliations:** Center for Biomedical Optics and Photonics and College of Physics and Optoelectronic Engineering, Key Laboratory of Optoelectronic Devices and Systems of Ministry of Education and Guangdong Province, Shenzhen University, Shenzhen 518060, China; zrl1197840542@163.com

**Keywords:** rheumatoid arthritis, osteoarthritis, photobiomodulation, animal arthritis models, anti-inflammation, clinic trials

## Abstract

Rheumatoid arthritis (RA) and osteoarthritis (OA) have a significant impact on the quality of life of patients around the world, causing significant pain and disability. Furthermore, the drugs used to treat these conditions frequently have side effects that add to the patient’s burden. Photobiomodulation (PBM) has emerged as a promising treatment approach in recent years. PBM effectively reduces inflammation by utilizing near-infrared light emitted by lasers or LEDs. In contrast to photothermal effects, PBM causes a photobiological response in cells, which regulates their functional response to light and reduces inflammation. PBM’s anti-inflammatory properties and beneficial effects in arthritis treatment have been reported in numerous studies, including animal experiments and clinical trials. PBM’s effectiveness in arthritis treatment has been extensively researched in arthritis-specific cells. Despite the positive results of PBM treatment, questions about specific parameters such as wavelength, dose, power density, irradiation time, and treatment site remain. The goal of this comprehensive review is to systematically summarize the mechanisms of PBM in arthritis treatment, the development of animal arthritis models, and the anti-inflammatory and joint function recovery effects seen in these models. The review also goes over the evaluation methods used in clinical trials. Overall, this review provides valuable insights for researchers investigating PBM treatment for arthritis, providing important references for parameters, model techniques, and evaluation methods in future studies.

## 1. Introduction

The two most common types of arthritis are osteoarthritis (OA) and rheumatoid arthritis (RA). OA is a common degenerative joint disease that is characterized by progressive and uneven loss of articular cartilage, bone spurs, and hardening of the underlying bone, as well as a variety of abnormalities in the synovium and around the joint. Aging, obesity, genetics, and previous joint injuries are all risk factors for OA, which primarily affects women [1,2].

RA, on the other hand, is a chronic autoimmune disease that primarily affects the elderly, with women being more affected than men. RA patients’ immune systems attack healthy synovial joints, resulting in continuous polyarticular synovitis, cartilage and bone damage, and joint failure, which causes severe pain, swelling, and fever. As the disease progresses, it can cause disability, reducing the patient’s quality of life and increasing the financial burden [3,4].

Unfortunately, there are currently no effective treatments for RA and OA that are free of side effects. Nonsteroidal anti-inflammatory drugs and steroids (NSAIDs) are commonly used by experts to treat pain and joint stiffness caused by inflammation. Although these medications effectively relieve RA symptoms, they do not provide a long-term cure. The European League Against Rheumatism proposed the use of disease-modifying anti-rheumatic drugs (DMARDs) to treat RA in 2010 [5]. DMARDs take time to work; they can significantly slow down disease progression, prevent the development of RA, and improve joint deformity. However, all current drug treatments for RA and OA have negative side effects, and long-term use of these drugs can put financial strain on patients [6]. As life expectancy rises, so does the number of elderly patients with arthritis, creating an urgent need for effective, side-effect-free treatments for RA and OA.

Photobiomodulation (PBM) has shown promising results in the treatment of neurodegenerative diseases, burns, wounds, and trauma in recent years by exerting anti-inflammatory, antioxidative, and ion channel-regulating effects [7,8,9,10]. PBM uses red or near-infrared light to activate cytochrome C oxidase in mitochondria, resulting in a variety of biological responses. Low-dose laser or LED light has been shown to reduce inflammation, increase ATP production, and regulate enzyme and gene expression in PBM patients [11,12]. The anti-inflammatory properties of PBM provide a theoretical foundation for its potential to alleviate arthritis-associated symptoms [13]. Notably, PBM does not require heat to be effective, and temperature changes during light irradiation are minimal [14,15]. Despite promising results in the treatment of a variety of clinical diseases, including neurodegenerative diseases and alopecia [16,17], the efficacy of PBM remains debatable, and research on PBM parameters is inconsistent. The efficacy of PBM treatments for RA, in particular, is still being debated. While some studies have shown that PBM can reduce inflammation and repair cartilage in arthritis [18,19], others have failed to find significant differences between PBM treatment and placebo groups [20,21,22]. Disparities in PBM efficacy may be explained by differences in PBM parameters such as wavelength, power density, light dose, and treatment duration.

In this review, “Low-level laser therapy”, “rheumatoid arthritis disease”, and “photobiomodulation” keywords were searched in Google Scholar, PubMed, and Medline. And the search for studies on PBM for arthritis ranged from 1987 to 2022.

This review provides an in-depth look at the potential of PBM in arthritis treatment. It covers three levels of investigation: cellular mechanisms, small animal models, and clinical studies. In terms of cellular mechanisms, the review describes how PBM can regulate arthritis-related cells by reducing inflammation, promoting tissue repair, and influencing cellular metabolism and signaling pathways. Moving on to small animal arthritis models, the review emphasizes the importance of quantitative and effective research in understanding the potential benefits of PBM for arthritis treatment. The review then delves into clinical studies of PBM in the treatment of arthritis, providing an accurate summary of the methods used to evaluate arthritis and the outcomes of PBM in clinical settings. Overall, this systematic and informative review is a valuable resource for those interested in learning more about PBM’s applications in arthritis treatment.

## 2. Arthritis

### 2.1. Joint

Joints, which are made up of two or more bones, play an important role in facilitating movement within the body. A synovial membrane surrounds movable joints, providing lubrication and nourishment to the joint tissues, including the cartilage. The presence of articular cartilage, a smooth and hard covering on the ends of bones, ensures that joints move smoothly and uninhibitedly as long as the cartilage is free of damage or lesions [23,24]. Joints not only allow for easy and precise movement, but they also support the weight of the body. However, arthritis severely impairs joint function and interferes with daily activities. Pain, swelling, heat, and inflammation are all symptoms of arthritis [25]. RA and OA are the two most common types of arthritis, and they differ in their pathological mechanisms, evaluation, and treatment methods.

### 2.2. RA

RA is a joint disease that primarily affects the wrists, hands, knees, and ankles. Most of the time, these joints are affected symmetrically [26]. However, the etiology and pathogenesis of RA are complicated, and the precise pathological mechanism is unknown. RA is an autoimmune disease that occurs when the immune system incorrectly attacks joint and organ tissues. Leukocytes and monocytes infiltrate the joint synovium and release cytokines that attack fibroblast-like synoviocytes in RA (FLS). As a result, inflammatory factors are released, causing new blood vessels to form in the synovium. This causes the synovial cavity on both sides to thicken, resulting in the formation of a pannus [27]. The pannus grows and invades the joints over time, causing cartilage and bone destruction [28].

Lymphocytes, white blood cells, monocytes, and macrophages are among the cells that infiltrate the synovial cavity during RA inflammation. These cells release inflammatory mediators that cause joint inflammation, resulting in fluid accumulation and joint swelling [29,30,31]. Furthermore, the swelling puts pressure on the surrounding nerves, causing pain. Inflammatory mediators also play a role in the subsequent stages of the inflammatory response, initiating a cascade of inflammatory effects in different cells that stimulates FLS proliferation, eventually leading to cartilage and bone destruction [28]. As RA progresses, joint inflammation causes a narrowing of the gap between joints, resulting in joint stiffness and reduced mobility, which is also responsible for the RA-specific morning stiffness [32]. The bones between the joints may fuse together in advanced stages of RA, rendering the joints immobile.

RA is a complex disease that is influenced by both genetic and environmental factors. The presence of anti-citrullinated protein antibodies (ACPAs), which are produced by B lymphocytes, divides RA patients into two subtypes. ACPA is a highly specific biomarker used in clinical trials to diagnose and predict the onset of RA. Approximately 67 percent of RA patients test positive for ACPA, and those who are ACPA-positive generally have more severe symptoms than those who are ACPA-negative [32,33,34]. Both genetic and environmental factors influence ACPA production. HLA-DR1 and HLA-DR4 have been identified as the genes most strongly associated with ACPA-positive RA in studies [35]. Furthermore, environmental factors such as smoking and dust exposure can activate immune cells, triggering immune responses that contribute to the development of RA. Furthermore, the gut microbiota has been linked to the pathogenesis of RA via various molecular mechanisms [36,37]. T lymphocytes and B cells become activated in response to genetic and environmental cues, resulting in the production of ACPA and, eventually, the onset of RA. Monocytes can become active and differentiate into macrophages during joint inflammation, with either a pro-inflammatory (M1) or anti-inflammatory (M2) phenotype. According to research, ACPA-negative patients have a higher proportion of the M1/M2 cell phenotype than ACPA-positive patients [38]. This finding suggests that targeting the regulation of macrophage phenotype could be a promising treatment strategy for RA.

External manifestations of synovitis include joint swelling and pain as a result of immune activation. A multitude of immune cells infiltrate the synovial cavity, including B lymphocytes (humoral immunity), T lymphocytes (cell-mediated immunity), monocytes, macrophages, mast cells, dendritic cells, and fibroblast-like synoviocytes (FLS), producing inflammatory factors such as TNF-α and IL-1β [39]. ACPA has been shown to activate NF-kB and increase TNF-α production in macrophages, resulting in a more severe clinical phenotype in ACPA-positive RA patients [40]. These cells require a constant supply of nutrients and oxygen, and the primary pathological process in RA, pannus, provides a supportive environment for synovial lining proliferation and subsequent bone invasion [41,42]. Pannus also causes the formation of new blood vessels, which supply nutrients and oxygen to inflammatory cells, thereby promoting the persistence of RA. VEGF, a potent endothelial cell-specific growth factor, is upregulated by pro-inflammatory cytokines and hypoxia and is synthesized by a variety of cells, including macrophages and FLS. Its concentration is generally high in the serum of RA patients [43]. Inhibiting pannus vascularization is a potential therapeutic strategy for RA blood vessel targeting.

To manage RA, doctors typically use a combination of pharmacological treatments, with the goal of reducing inflammation, relieving pain, and slowing cartilage damage. Nonsteroidal anti-inflammatory drugs (NSAIDs) [44], steroids, and standard disease-modifying antirheumatic drugs (DMARDs) [45] are among the treatment options. When standard DMARDs are ineffective, doctors may prescribe biologic DMARDs [46]. Physical therapy is also frequently recommended as part of the RA treatment regimen to maintain joint flexibility and muscle strength [47]. In severe cases of RA that do not respond to the aforementioned treatments, surgical interventions may be considered. For severely damaged joints, joint replacement surgery, such as total joint replacement, may be performed [48]. When joint replacement is not an option, arthrodesis may be considered, which involves removing the damaged joint and fusing it with pre-grown bone. Another treatment option for severe RA is synovectomy, which involves removing the synovium surrounding the joint and replacing it with artificial joints [49].

### 2.3. OA

OA is the most common type of arthritis, and its prevalence rises with age, especially in female patients and in weight-bearing joints [50]. The most common type of OA is knee OA. The knee joint is made up of the femur, tibia, and synovial membrane, with articular cartilage covering the end of the femur. The cartilage and bone in a healthy knee joint are smooth and without folds, and the synovial fluid in the synovial cavity acts as a lubricant, allowing painless motion with minimal friction and contact between the upper and lower bones of the knee [51]. OA, on the other hand, affects the cartilage and synovial fluid in the knee joint. Although the joint space appears normal in mild OA, the cartilage matrix has begun to decompose, and dense bone spurs form on the cartilage edge. Moderate exercise and weight loss can help reduce the load on the knee bone and joint, alleviating OA symptoms [52].

Changes in the joints become more noticeable in the middle stage of OA development, and the surface between the bones begins to erode. The cartilage, which is important for lubrication and protection, is also significantly degraded and worn out, resulting in a reduction in joint space. The viscosity and lubricating properties of synovial fluid deteriorate as well. OA typically affects the subchondral bone [53,54]. When the subchondral bone wears flat, oxygen enters the cartilage to try to heal itself. Dendritic cells and lymphocytes release cytokines and nuclear proteins into the synovial fluid, causing an inflammatory environment to form. The size and number of bone spurs may increase at this stage, roughening the bones and causing more severe and persistent joint pain. OA can be treated with pain relievers and steroid medications, as well as moderate exercise and weight loss.

The joint space becomes increasingly narrow in severe OA, resulting in rapid and severe cartilage and bone degradation and wear, a decrease in synovial fluid, inflammation of the knee joint, increased pain, and limited mobility. Inflammatory cells release damaging proteins and cytokines that degrade the cartilage and soft tissue surrounding osteophytes, resulting in an increase in osteophytes and direct contact between the upper and lower bones of the joint. At this point, the joint has lost its ability to move and bear weight, and surgical replacement of a portion or the entire joint is the only viable treatment option [53,55].

## 3. The Mechanism of PBM on Arthritis Treatment

PBM promotes cell activity and functional normalization by regulating a number of cellular responses, including promoting mitochondrial ATP production, releasing intracellular nitric oxide (NO), and regulating immune cell secretion of inflammatory cytokines such as TNFα, IL-6, and IL-β. Furthermore, the regulation of enzymes and genes is critical. Figure 1 depicts the sequence of events that take place when light is applied.

### 3.1. ATP

According to research, photons emitted by PBM are primarily absorbed by cytochrome C oxidase (CCO) within the fourth chain of the mitochondrial electron transport chain, resulting in a series of complex cellular responses and altered redox states [56]. Two primary redox processes can explain these reactions. First, photoexcitation of specific chromophores within CCO causes changes in their redox properties, which then speeds up electron transfer. Second, upon photo-dissociation, CCO releases NO, increasing oxygen binding and respiration rates [57]. Furthermore, both CCO and NADH flavoproteins function as photoreceptors, causing changes in cytoplasmic proton motility, mitochondrial transmembrane potential, pH, and cellular redox potential by rapidly activating the mitochondrial respiratory chain and NADH oxidation pool [58].

When cells are exposed to red or near-infrared light, CCO absorbs photon energy, causing an electronic transition from a low-energy level to a high-energy level on the chromophore, resulting in the release of an electron that participates in cell respiration and ATP synthesis. As a result, PBM can improve cellular respiration efficiency by promoting cellular metabolism and increasing cell membrane potential [59]. Increased cellular energy levels can contribute to better cellular states, such as cell proliferation and normal functional cell activity.

### 3.2. Blood Flow

Bone is a highly vascularized tissue, and blood supply is crucial in bone reconstruction [60,61]. It is well known that increased blood flow to bone tissue promotes bone regeneration. PBM has been shown in previous studies to increase blood flow [62]. When photons are absorbed by cells, they directly photo-dissociate NO in the cell’s mitochondria, allowing NO to easily cross the cell membrane and stimulate smooth muscle cells in the inner wall of blood and lymphatic vessels, causing vasodilation and increased blood circulation [63]. Poor blood flow in the affected area of arthritis patients can cause nerve compression and pain. PBM, on the other hand, can regulate normal blood flow, reducing pain and promoting healing. Furthermore, the space previously occupied by NO in cells can be replaced by O_2_, providing raw materials for cell respiration and facilitating ATP production [64]. Notably, the effect of PBM on NO release regulation is transient, as NO release in CCO stops immediately when the light source is removed [63,65,66]. Additionally, Tim et al. [67] found that PBM could regulate the expression of genes involved in inflammation and angiogenesis. Using real-time PCR to detect inflammation and angiogenesis genes in rats, the results showed that angiogenesis genes were significantly upregulated after 36 and 72 h of PBM treatment, thereby stimulating angiogenesis. Taken together, PBM can regulate blood flow by releasing intracellular NO and modulating the expression of angiogenesis genes.

### 3.3. Regulation of Cytokines

Various cytokines, including interleukins, activate specific cells in arthritis, regulating or hastening inflammatory processes by activating transcription factors [68]. Cytokines are a class of molecules, mainly produced by T cells, macrophages, and endothelial cells, which mediate the activation of cells during immune or inflammatory reactions. Major regulators of the inflammatory response include TNF-α and interleukins, with changes in IL-6 and IL-1β levels commonly used as indicators to evaluate treatment effects [69,70,71,72]. The release of these cytokines can influence the secretion of other cytokines, creating a cascade effect. For instance, TNF-α not only mediates inflammation, immune processes, and proteolysis but also stimulates the production of IL-6 in the cytokine secretion cascade. Excessive TNF-α secretion in RA’s inflammatory synovium causes abnormal immune responses of T cells [73]. While transient synthesis of IL-6 induces beneficial responses to infections and tissue damage, continuous secretion can lead to autoimmune diseases [74]. Knockdown of the IL-1β gene expression significantly reduces the inflammatory response in some autoimmune diseases like RA [75]. Studies on RA treatment with PBM mainly focus on these cytokines to verify therapeutic effects. Different PBM parameters yield significantly different results. For example, in an OA mouse model, Alves et al. found that using 50 mW/cm^2^ laser power density (808 nm) was more effective than 100 mW/cm^2^ in reducing inflammation, with more significant downregulation of IL-1β and IL-6. However, at 100 mW/cm^2^ power density, the downregulation of TNF-α was more prominent. These findings demonstrate that different PBM parameters have varying anti-inflammatory effects, most directly reflected in the production of inflammatory cytokines [69,70,71,72,76,77,78]. In clinical experiments, PBM inhibits the production of inflammatory factors. For instance, Adly et al. measured the level of IL-6 in patients after laser treatment and found that it was significantly reduced after laser irradiation [79]. Additionally, PBM upregulates anti-inflammatory factors, such as transforming growth factors-β (TGF-β) [80]. TGF-β, produced by synovial fibroblasts, inhibits the production of TNF-α and other inflammatory factors [81]. Bartoli et al. [82] reported increased TGF-β content in a Wistar mouse model of arthritis irradiated with 670 nm lasers. Overall, PBM plays a regulatory role in arthritis by reducing the production of pro-inflammatory cytokines and upregulating anti-inflammatory cytokines [83].

### 3.4. Enzyme

The primary cause of pain in RA patients is the release of inflammatory substances by cells in response to local inflammation, which stimulates the transduction of cell signals to produce pain-causing substances, such as the relative enzyme. Prostaglandin E2 (PGE2) is a key player in the pain-inducing process and has become the primary representative enzyme of inflammation-related pain [84]. NSAIDs and selective cyclooxygenase 2 (COX-2) inhibitors are commonly used to alleviate RA-related swelling and pain by effectively inhibiting COX-2 activity and reducing PGE2 synthesis [85,86]. PBM also exhibits an anti-inflammatory effect by inhibiting COX-2. Lim et al. reported that after 635 nm LED light irradiation of human gingival fibroblasts, the decrease in reactive oxygen species (ROS) in cells inhibited COX-2 production. Additionally, in animal experiments involving tibial deprivation, COX-2 content tended to be downregulated after 7 days of PBM treatment [67].

In RA patients, IL-1β and TNF-α can stimulate matrix metalloproteinase (MMP-13), which degrades all components of the extracellular matrix, including cartilage in joints. MMPs-1 also plays a crucial role in cartilage degradation in both RA and OA. Moreover, other MMPs, such as MMP-2 and MMP-3, are expressed in arthritic joints, and their content is increased [87]. Therefore, inhibiting MMP production is vital for preventing cartilage damage. A recent study demonstrated that irradiation with an 808 nm laser at 50 J/cm^2^ can significantly reduce the contents of COX-1 and MMP-13 in arthritic mice [77]. PBM plays a critical role in regulating the content and viability of enzymes associated with arthritis, protecting articular cartilage, and reducing the inflammatory effects of arthritis. In a word, PBM effectively reduces COX-2 content, leading to the inhibition of PGE2 production, relieving pain, and swelling in joints. Additionally, it inhibits metalloprotein enzymes, alleviating the erosion and degradation of joint tissues and components, preserving the normal function of joints.

### 3.5. Gene

Gene expression regulates many cellular functions, including protein secretion and signaling factor expression. As a result, it is critical to investigate whether PBM can modulate gene expression. In arthritis, inflammation is caused by the infiltration of cells such as single cells, macrophages, lymphocytes, and granulocytes into the synovial cavity, with polymorphonuclear (PMN) leukocytes being key factors that contribute to joint injury [88]. PBM has been shown to increase the expression of both anti-apoptotic (p53 gene) and pro-apoptotic (Bcl2) genes in PMN leukocytes, inducing apoptosis in PMN cells at higher PBM concentrations [89]. Additionally, PBM has shown the ability to regulate the inflammatory process, promoting early granulation tissue deposition and new bone tissue in bone injury areas. It upregulates several proinflammatory and pannus genes such as FGF14, FGF2, ANGPT2, ANGPT4, and PDGFD after 36 and 72 h of PBM. Furthermore, PBM regulates the expression of inflammatory and angiogenic genes and the immune expression of COX-2 and VEGF at the initial stage of bone healing, contributing to new bone development [89]. Inflammation is also caused by the infiltration of inflammatory cells from blood vessels into the synovial cavity. CXCR4, a chemokine receptor involved in various inflammatory diseases mediated by peripheral blood immune cells, mediates the abnormal penetration of immune cells into joint and lung tissues, resulting in lymphocytic infiltration in the synovial cavity of the joints. Moreover, the entry of inflammatory cells from blood vessels into the synovial cavity is also a key step causing inflammation. Therein, the chemokine receptor CXCR4 is involved in various inflammatory diseases mediated by peripheral blood immune cells. Studies have shown that CXCR4 and its ligand CXCL12 mediate the abnormal penetration of immune cells into joint and lung tissues [90]. PBM using 830 nm GaAlAs diode irradiation has been found to downregulate the expression of CXCR4 mRNA, which may be one of the mechanisms of PBM inhibiting RA inflammation [91]. Other studies have also shown that PBM can regulate the expression of CCL2 and type II collagen [72,78]. Investigating the gene regulation of PBM is crucial for understanding its mechanism in treating RA, as it involves changes in protein expression and cellular function, which are critical steps in exploring the mechanism of PBM treatment.

## 4. PBM Regulates Arthritis-Related Cells

This section examines the regulatory effects of PBM on key cells involved in RA pathogenesis, such as macrophages, PMN cells, T cells, and FLS. We gain valuable insights into the therapeutic potential of PBM for RA treatment by elucidating the molecular-level modulation of these cells by PBM. This knowledge could pave the way for more targeted and effective PBM-based therapies, advancing RA management and improving patient outcomes.

### 4.1. Polymorphonuclear Cells

A variety of chemokines and cytokines activate PMN cells, causing them to migrate from the blood vessels to the lubricating membrane compartment in the synovial cavity. Once there, they release a variety of pro-inflammatory cytokines, causing inflammatory cell proliferation, pannus formation, increased vascular permeability, and stimulation of synovial stromal cells and cartilage cells to secrete proteolytic enzymes (MMP-2, -9, and -13). These enzymes promote the degradation of type II collagen and alter the glycan composition and water storage capacity of articular cartilage, resulting in physiological changes and joint functional destruction [92]. As a result, inhibiting the proliferation and viability of PMN cells is an effective way to treat RA, and there are related studies on PBM regulating the function of PMN cells. For example, one study found that higher light doses (30 J/cm^2^) with an 830 nm laser increased pro-apoptotic gene expression, while lower doses (3 J/cm^2^) decreased it. These findings suggest that the biochemical reactions differ significantly depending on the light dose [89]. Furthermore, other laser wavelengths inhibit PMN proliferation. Carlos et al. used 660 nm lasers at a low light dose (2.5 J/cm^2^) to treat zymosan-induced arthritis mice, and the results showed that the amount of PMN in the synovium of joints was significantly reduced [93]. The levels of pro-inflammatory cytokines such as IL-1β and IL-6 decreased as well. PBM may have a more specific effect on PMN cells due to the additional mechanism of free radical production by these cells during inflammation and their very short half-life, accelerating cellular functions such as cytokine production, which leads to apoptosis of PMN cells [89,94]. As a result, in the treatment of RA, PBM may reduce joint inflammation by inhibiting PMN viability and proliferation.

### 4.2. Macrophages

Macrophages play a crucial role in the pathogenesis of RA through antigen presentation, osteoclast generation viability, and secretion of proinflammatory cytokines such as TNF-α, IL-1, and IL-6 [95]. TNF-α promotes the expression of other cytokines, while IL-1β induces the release of adenosine from monocytes, activates PMN and oxidative stress, and increases the release of cytokines and chemokines from synovial fibroblasts. IL-1β and TNF-α can also activate the release and production of MMPs. On the other hand, IL-6 facilitates the proliferation of T and B cells, antibody production, hematopoiesis, and platelet formation [96,97]. It is noteworthy that macrophages have two phenotypes with completely different functions in response to inflammation. For example, M1 macrophages regulate inflammation and cell proliferation during muscle repair, while M2 macrophages guide differentiation and remodeling and promote tissue regeneration. However, 780 nm and 660 nm lasers (1 J/cm^2^) can effectively reduce the M1 phenotype macrophages while increasing the expression of M2 phenotype and reducing the content of proinflammatory cytokines [98]. Therefore, exploring the regulation of macrophage phenotype by PBM is also a promising approach for the treatment of RA. Different light power densities also have different regulatory effects. Alves et al. irradiated OA mice with 808 nm laser at 50 mW/cm^2^ and 100 mW/cm^2^ power density, respectively. The results showed that 50 mW/cm^2^ laser inhibited the secretion of IL-1β and IL-6 in macrophages more effectively. A power density of 100 mW/cm^2^ was more effective in reducing TNF-α production, suggesting that PBM can directly regulate inflammatory cytokine secretion of macrophages in addition to regulating macrophage phenotypes [76].

### 4.3. T (Treg) Cells

As RA is a heterogeneous autoimmune disease, immune lymphocytes also play a significant role in its pathogenesis. The infiltration of T cells into the synovial fluid is an important cause of inflammation, including Th17 cells that secrete proinflammatory factors like IL-17 [99]. The immunosuppressive activity of regulatory T (Treg) cells can modulate joint inflammation [100]. The mechanism of T cell immunosuppression involves the secretion of anti-inflammatory cytokines such as TGF-β and IL-10. Additionally, some inflammatory mediators directly activate the secretion of perforin and activate granzyme A, which are common molecules in CD8+ cytotoxic T cells, leading to the inhibition of molecular effects associated with immunosuppression by effector T cells. Moreover, T cells can inhibit the maturation of DC cells, promote the downregulation of CD80/CD86 expression, and compete with effector CD4+ cells for interaction with antigen-capturing and antigen-presenting cells [101].

T cells are mainly differentiated into two phenotypes, namely CD4+ and CD8+ T lymphocytes. Macrophages and DC cells activate T cells through the presentation of CD80/CD86 and histocompatibility complex (MHC) antigens, leading to differentiation into the aforementioned two phenotypes [102]. An increase in the number of macrophages and DCs results in more antigen presentation in lymph nodes, providing a favorable cytokine environment for the differentiation and proliferation of T lymphocytes. CD4+ T cells contribute to a positive feedback loop in the chronic inflammatory response due to the production of cytokines and chemokines. Furthermore, aggravated inflammation stimulates macrophages and DCs to migrate to the inflammatory site, where they present antigens to stimulate the differentiation of CD4+ lymphocytes, thereby exacerbating chronic inflammation [103].

CD8+ T cells demonstrate two opposite responses under different stimulations in the immune response. On the one hand, they maintain the chronic inflammatory process by secreting high levels of proinflammatory cytokines and lysozymes; on the other hand, they can inhibit inflammatory responses in arthritis by secreting IL-10, an anti-inflammatory cytokine [104]. Therefore, the increase of CD8+ lymphocytes can contribute to inhibiting the occurrence of inflammation. A study has demonstrated that an 830 nm laser at a high dose (30 J/cm^2^) upregulates the expression of CD25, the receptor for IL-2 on the cell surface, and CD8+ T cells with high CD25 expression facilitate the feedback of negative immune responses in the inflammatory microenvironment, leading to the alleviation of inflammation [89].

### 4.4. Fibroblast-like Synoviocytes

FLS are a crucial cellular component of joint synovial tissue, and there exists a complex network of FLS–macrophage–lymphocyte interactions in the inflammatory system of RA. Despite the fact that numerous inflammatory cells infiltrate the synovial cavity, the primary cause of synovial hyperplasia is the excessive proliferation and activation of FLS [105]. Additionally, FLS produces a plethora of pro-angiogenic factors, forming new blood vessels to provide nutrients and oxygen for inflammatory cells [42]. Thus, a decrease in FLS directly inhibits the inflammatory responses. Hsieh et al. [106] irradiated arthritic mice with 780 nm GaAlAs lasers at 4.5 J/cm^2^ to explore the effect of PBM on the activity and number of nuclear FLS cells. The results showed that PBM effectively reduced the number of FLS in the affected site, as well as the levels of inflammatory mediators such as TNF-α and MMP.

PBM, as shown in Figure 2, slows arthritis primarily through the following activities. On the one hand, PBM regulates macrophage phenotypes [107,108], increases the production of cytokines such as IL-10 and TGF-β, promotes tissue repair, and reduces the content of inflammatory factors such as TNF-α, IL-1β, and IL-5. This, in turn, inhibits the differentiation of CD4+ T cells [70] and reduces the content of TNF-α, IL-1β, IL-5, and other substances, forming benign, negative anti-inflammatory feedback. On the other hand, PBM reduces the apoptosis of PMN and FLS cells and reduces the infiltration of these inflammatory cells in arthritis joints [89,109,110]. Furthermore, the effects of PMN-mediated oxidative stress and vascular permeability will be inhibited [111,112,113], and VEGF secretion of FLS will also be reduced, inhibiting the cardiovascular production that provides nutrients for infiltrated inflammatory cells. Therefore, PBM slows down the formation of pannus and reduces the destruction of joint cartilage.

## 5. PBM Regulates Arthritis Animals

### 5.1. Establish Arthritis Animal Models

The establishment of an effective animal model of arthritis is fundamental for investigating the modulatory effects of PBM on arthritis. Various methods can be used to induce arthritis in mice, such as collagenase-induced arthritis (CIA), Complete Freund’s Adjuvant (CFA), zymosan induction, papain solution-induced arthritis, microcrystalline arthritis (induced by physiological solution, hydroxyapatite, and urate solution), and surgical intervention. These methods induce arthritis through different mechanisms, ultimately resulting in inflammation and structural and functional destruction of arthritic joints. Table 1 presents a summary of various methods utilized for the induction of arthritis, along with their corresponding mechanisms. This table serves as a useful reference for researchers investigating arthritis induction in animal models.

Among these methods, CIA is a common and effective way to mimic human RA in mouse models. It is widely used to investigate the pathological mechanisms of RA and search for effective treatment approaches [114]. The pathological characteristics of mouse models induced by CIA typically include proliferative synovitis with infiltration of polymorphonuclear and mononuclear cells, pannus formation, cartilage degeneration, and bone erosion [115].

Another common method to establish animal models of arthritis is through the use of CFA solution induction. In 1963, Pearson et al. found that immunizing rats with CFA solution could induce arthritis. In this method, T cells have an immune response to autologous components induced by the adjuvant. However, this animal model is different from real arthritis in terms of reorganization and immunological characteristics. It is typically used to study the mechanism of autoimmune responses induced by external factors [116].

The mechanism by which zymosan solution induces arthritis in animals is by stimulating the secretion of lysosomal enzymes through the activation of macrophages, which induces inflammatory responses in the joints. This is a process mediated by cell surface receptors. During the induction phase of the immune response, phagocytes (monocytes, macrophages, and dendritic cells) bind zymosan with receptors and activate NF-kB channels, leading to the production of inflammatory factors and the expression of co-stimulating molecular CD80. Furthermore, zymosan induces DC maturation and IL-2 production to trigger adaptive immune responses, whereby mature DCs migrate to lymph nodes and induce the activation of T cells, which proliferate via antigen presentation. In the provocation phase, T lymphocytes stimulated by zymosan are recruited and produce a variety of cytokines, leading to the amplification of the inflammatory response into a more intense process [117,118,119]. Zymosan injection also leads to increased vascular permeability and edema, which are important events in maintaining the normal function of inflammatory cells and the source of plasma [120]. Injection of zymosan into mice resulted in inflammatory arthritis with monocyte infiltration, synovial hypertrophy, and pannus formation [119].

Upon injecting papain solution, a common method to establish animal models of osteoarthritis (OA), articular cartilage degeneration is primarily induced [121,122]. The initial symptom is characterized by the elevation and loosening of the cartilage surface, followed by thinning and fibrotic changes. Ultimately, cartilage breakdown and loss near the subchondral bone occur, closely resembling the clinical features of degenerative arthritis [123].

Microcrystal deposition is a common cause of joint diseases that often results in severe pain and inflammation. The main pathogenic agent is sodium urate crystals, responsible for gout, although calcium pyrophosphate can also be deposited in various clinical forms. During the pathogenesis of microcrystal-induced arthritis, a plethora of inflammatory cytokines, such as PGE2 and IL-1, are released [124]. To induce the formation of microcrystalline arthritis, hydroxyapatite and urate are typically injected into animal joints [125].

Transection of joint ligaments is also commonly used to model OA. This procedure stimulates the production of several cytokines, including TGF-β1, IL-1β, MMP-3, and TNF-α, in the subchondral bone [71,126]. Elevated levels of TGF-β1 lead to the formation of osteoid islets in the bone marrow and increased angiogenesis. Similarly, high levels of TGF-β1 have been observed in the subchondral bone of human OA patients [127]. Furthermore, other cytokines contribute to inflammatory responses, exacerbating subarticular osteochondral erosion and bone spur formation.

A recent study proposed an intriguing method to induce arthritis in mice by injecting small amounts of senescent cells into the knee joint [128]. Through the expression of luciferase, researchers could track the activity of senescent cells in the mice, and imaging and histology indicated the development of osteoarthritis. Notably, fibrocytes and mesenchymal stem cells have the potential to differentiate into senescence-related secretory phenotypes, producing inflammatory cytokines, chemokines, and growth factors [129,130]. This unique approach sheds light on the role of senescent cells in arthritis development and adds a novel dimension to arthritis research in animal models.

### 5.2. Studies on In Vivo Treatment of Arthritis with PBM

The biochemical simplicity of cell environments makes them suitable for investigating the mechanisms of PBM. However, it is crucial to validate whether PBM can effectively regulate arthritis at the in vivo level. Mouse models are widely preferred for arthritis research due to their cost-effectiveness, easy management, and high genetic similarity to humans [131,132], which saves resources while ensuring reliable results. The CD57BL/6 mice strain is particularly common in arthritis research, and most in vivo studies are based on this model. Additionally, rabbits can also serve as research models, given their larger size and distinguishable joint structures [78]. In animal experiments, researchers induce arthritis and administer PBM treatment before euthanizing the animals to collect joint lavage fluid for analysis of cell types, cell numbers, cytokines, and RNA components. Table 2 provides an overview of studies conducted on animal models to investigate the efficacy of PBM in treating arthritis. The table includes information on the type of animal model used, the parameters of PBM treatment, and the study results. By summarizing the available literature on animal studies, this table serves as a valuable resource for researchers interested in exploring the potential of PBM therapy for arthritis treatment.

Measuring changes in cytokines and inflammatory mediators in mouse joint lavage fluid before and after PBM is the most direct method to evaluate the effect of PBM on arthritis. Commonly measured mediators include MMP, IL cytokines, TNF-α, and VEGF. Several studies have reported significant decreases in the contents of PGE-2, TNF-α, MMP, IL-1, and IL-6 after PBM treatment in mice [67,69,71,72,76,82,110,133,139]. These changes result in the non-activation of response cells involved in arthritis, such as macrophages and lymphocytes that mediate the inflammatory response. This process creates a positive feedback loop for arthritis treatment [143,144]. To understand the reasons behind the decline in these inflammatory mediators, one study found that the number of cells producing these mediators significantly decreased after PBM treatment. Additionally, the number of FLS cells exhibited different responses to higher (72 J/cm^2^) and lower (4.5 J/cm^2^) PBM doses, with the latter leading to significantly lower cell numbers and subsequently downregulated TNF-α levels [110]. Another study investigated the effects of different power densities of PBM (50 and 100 mW/cm^2^ power output) on arthritis treatment in a mouse model. The results showed that the low power density had a more pronounced inhibitory effect on the number of macrophages and neutrophils, as well as the content of IL-6 and IL-1β [76]. However, due to different experimental conditions, such as laser types and equipment, the definition of low power is not clear, and no consensus has been reached.

In animal models, the therapeutic effects of PBM can be evaluated from various perspectives. Assessing angiogenesis, vascular permeability, and articular cartilage protection at the tissue and organ levels is not feasible with cell experiments alone. These arthritis-related symptoms are also targeted for treatment in clinical studies. As a result, some studies have investigated PBM’s therapeutic effects in small animal model species based on the joint’s structural organization and function. For instance, studies using 808 nm and 904 nm lasers on Zymosan- and CFA-induced arthritis mouse models have demonstrated that PBM can reduce angiogenesis, fibrotic tissue production, and protect articular cartilage properties [122,145]. Such evaluations provide valuable insights into the potential therapeutic benefits of PBM for arthritis treatment.

### 5.3. Effects of PBM Parameters on Animal Models of Arthritis

According to the studies summarized in Table 2, the choice of wavelength and power (dose) in PBM is crucial. Different wavelengths have different modulating mechanisms. For instance, the absorption peak of CCO in mitochondria is located at 600–1000 nm [146], and green light has been found to regulate the selective transmittance of ion channels in cell membranes [147]. However, it appears that the coherence of the PBM light source is not critical. Both LEDs and lasers have been used in studies and have shown certain regulatory effects on arthritis [78,89,134,148,149].

Wavelength is a crucial parameter in PBM, and longer-wavelength infrared light appears to be more effective than red light in terms of anti-inflammatory effects. Morais et al. [134] reported that shorter-wavelength light (628 nm) had no effect on modulating edema and vascular permeability in zymosan-induced arthritis mice, whereas longer wavelengths (685 and 830 nm) demonstrated better performance. In the study by Kuboyama et al. [149], the therapeutic effects of 570 nm LED light and 940 nm light on arthritis in mice were compared, and the results showed that 940 nm light had a better swelling reduction effect and was also more effective in inhibiting proinflammatory factors such as IL-1β, IL-6, and MMP-3. Souza et al. also reported a more effective performance of 780 nm laser in reducing inflammatory cells and promoting muscle fibers than 660 nm laser [98].

Furthermore, researchers have explored the synergistic effects of using two or more wavelengths of light to treat arthritis based on the different modulation effects of each wavelength. For example, Oshima et al. used alternating irradiation of 630 nm (red light) and 870 nm (infrared light) at high frequencies to target specific regulatory sites in arthritis mice. This approach resulted in the inhibition of collagen degradation and a corresponding decrease in inflammation at the joint site [78]. These findings highlight the importance of carefully selecting and matching light wavelengths to achieve the desired therapeutic effect in PBM treatment for arthritis.

On the other hand, it is important to consider that the experimental conditions and specific mouse models used in different studies varied significantly, as shown in Table 2, with a wide range of power densities employed, ranging from single digits to thousands of mW/cm^2^. This inconsistency in experimental parameters may contribute to the disparate conclusions drawn by various studies. Therefore, the selection of appropriate power density and dose is crucial in PBM research. It is essential to strike a balance between using a high-enough power density and dose to trigger the desired biochemical effects while avoiding tissue thermogenesis and ensuring that the power density and dose are not too low to achieve therapeutic benefits.

Several studies in animal models have indeed compared the effects of PBM at different power densities and doses, consistently indicating that lower power densities tend to be more effective in reducing inflammation and inducing apoptosis in proinflammatory cells [72,76]. However, the definition of “low dose” may vary among studies due to differences in equipment and experimental conditions, leading to potential inconsistency in the interpretation of results. To address this issue, it is recommended for researchers to conduct gradient experiments with varying power densities and doses during their investigations. This approach allows them to systematically evaluate the effects of PBM at different levels and identify the optimal dose based on the specific experimental outcomes. Subsequent experiments can then be conducted with the selected optimal dose to further investigate the effects of PBM in a more controlled manner. Overall, careful consideration and selection of appropriate power densities and doses in PBM experiments are crucial for obtaining meaningful and consistent results.

## 6. Clinical Studies of PBM in the Treatment of Arthritis

### 6.1. Clinical Evaluation of Arthritis

In clinical studies, invasive methods for arthritis evaluation should be avoided whenever possible because they are uncomfortable and may not be appropriate for all patients. Instead, a variety of non-invasive assessment methods, including digital approaches such as questionnaires and pain level assessments based on direct patient feedback, are used. Several questionnaires have been developed to assess arthritis symptoms and their impact on the lives of patients. These questionnaires include the visual analogue scale (VAS), the disability of the arm, shoulder, and hand (DASH), the health assessment questionnaire (HAQ), the Osteoarthritis Index (WOMAC) of Western Ontario and McMaster Universities, and the Saudi Knee Function Scale (SKFS) [110,150], rely on the patient’s subjective judgment through questionnaire responses to draw conclusions [22,151]. Each of these questionnaires uses a unique set of questions and conversion methods to assess various aspects of arthritis and its effects on patients. Given the cultural and linguistic differences among patients, questionnaires must be appropriately designed to ensure accurate and meaningful results. For example, the SKFS is tailored to Arabic patients, taking into account their language and comprehension patterns [150].

Among these assessment methods, the VAS is one of the most widely used tools for evaluating the effects of arthritis treatment. It involves a simple vernier ruler approximately 10 cm in length, with 10 scales marked on it, ranging from “0” (no pain) to “10” (most severe and unbearable pain). Patients rate their pain levels on this scale, providing a quantitative measure of pain severity [152]. Due to its simplicity and effectiveness, the VAS is a valuable tool for assessing pain responses in arthritis patients.

Objective evaluation methods are indeed essential in arthritis assessment, and they are often combined with subjective evaluations to provide a comprehensive understanding of the disease and its impact on patients.

One common objective evaluation method is to use a goniometer to measure the range of motion (ROM) of the affected joint, such as the knee joint. This technique evaluates the angle of motion and functional limitations of arthritis patients, providing important information about joint mobility and stiffness [153]. In addition, collecting peripheral blood from arthritis patients is another instructive method for diagnosis and monitoring disease progression. Rheumatoid factors and anti-cyclic citrullinated protein are important indicators for diagnosing RA [154,155]. However, it is essential to consider other clinical and laboratory findings for a confirmed diagnosis of RA, as the presence of these factors alone may not be conclusive. Inflammatory markers in peripheral blood are commonly measured to assess the level of inflammation in arthritis patients. Erythrocyte sedimentation rate (ESR) tests can indicate inflammation if sedimentation is rapid, providing an indication of ongoing inflammatory processes [156]. Additionally, the concentration of inflammatory factors such as C-reactive protein (CRP) tends to increase in response to inflammation events [157]. Using enzyme-linked immunosorbent assays (ELISA) and other methods, researchers and clinicians can measure changes in the levels of inflammatory factors in the peripheral blood serum of patients, including interleukin-6 (IL-6), NF-KB, and CRP [79].

Radiographic analysis, particularly X-ray computed tomography (X-CT), is indeed an essential and effective evaluation method for assessing joint health and structural changes in arthritis patients. X-ray scans can provide a clear picture of the joint space width (JSW) and structural integrity of the affected joints, allowing researchers and clinicians to monitor disease progression and treatment effects. In clinical studies, X-ray scans can be used to select appropriate patients for experimental trials, and researchers often exclude patients with severe genu varus or genu valgus based on X-ray analysis [151]. For example, Gopal et al. [158] conducted X-CT scans on patients with knee OA before and after laser treatment, and the results showed a significant increase in JSW after 8 weeks of laser irradiation compared to patients who did not receive laser treatment (laser: 4.2 ± 0.3, placebo laser: 2.8 ± 0.6). This objective assessment provides valuable quantitative data on the efficacy of PBM in promoting joint health.

Morning stiffness is another useful evaluation indicator in assessing the therapeutic effects of PBM in arthritis patients [19]. It is a common symptom experienced by arthritis patients and can be assessed subjectively to gauge treatment outcomes and improvements in joint function. The assessment methods used in clinical trials investigating the efficacy of PBM in arthritis treatment can be broadly categorized into objective and subjective judgments. Objective evaluations, such as radiographic analysis and laboratory measurements of inflammatory markers, provide scientific and quantitative data, while subjective evaluations, like patient questionnaires and pain level assessments, offer insights into patients’ experiences and perceptions of treatment effects. Table 3 provides a comprehensive overview of the evaluation methods used in clinical trials, outlining the underlying principles of each method. This table serves as a valuable resource for researchers and clinicians interested in utilizing these evaluation methods to obtain scientifically accurate and meaningful results in their own studies.

By combining both objective and subjective evaluation methods, researchers can gain a comprehensive understanding of the therapeutic effects of PBM in arthritis treatment, enabling them to make informed clinical decisions and develop effective treatment strategies for patients.

### 6.2. Clinic Trials

Clinical trials examining the therapeutic effects of PBM in the treatment of arthritis have yielded encouraging results, particularly in knee OA and RA. PBM has been shown to reduce pain, swelling, and morning stiffness, indicating its ability to suppress arthritic inflammation [79,151,158,159,160,161,162,163,164,165,166,167,168]. However, it is important to note that some studies have reported no or only partial effects of PBM on arthritis treatment, indicating the need for better-defined treatment parameters and evaluation methods [169,170,171]. For example, Meireles et al. conducted a study with a subset of 82 patients with hand RA who received laser therapy. However, despite using VAS, HAQ, and DASH assessments, they concluded that there was no significant difference between patients who received laser therapy and those who did not [22]. Similarly, other studies on OA have reached similar conclusions to Meireles et al., suggesting that PBM does not provide substantial relief from inflammation or pain [20,21,169,170]. Additionally, specific PBM parameters need to be better defined [172]. The efficacy of PBM remains controversial due to variations in operating methods, treatment parameters (such as wavelength, power density, light source type, and arthritis type), evaluation methods, and irradiation sites in each research project.

Most clinical studies employ wavelengths in the 650–1000 nm range, which corresponds to the “optical window” known to effectively regulate cellular activities [12]. Moreover, the selection of power density is relatively conservative, ranging from 20–3000 mW/cm^2^, to avoid excessive heat generation that could cause burn damage to the patient’s skin. Studies should consider adhering to the American National Standard Institute (ANSI) standards to select the appropriate optical power density threshold, preventing excessive light-induced heat that may lead to skin or tissue burns. As seen in Table 3, almost all studies use VAS to evaluate the degree of arthritis, as it is a mature and reliable detection method widely used in arthritis treatment evaluation.

By analyzing the data from these studies, researchers can gain a better understanding of the optimal PBM parameters and evaluation methods for arthritis treatment, leading to more standardized and effective clinical approaches. Additionally, further research is needed to address the inconsistencies and explore the potential mechanisms underlying the variable treatment responses, ultimately advancing the field of PBM and its applications in arthritis management.

Table 4 presents a comprehensive summary of studies investigating the efficacy of PBM in OA and RA treatment. It includes detailed information on the PBM parameters and evaluation methods used in each study, along with the final conclusions and effect analysis of each paper. This table serves as a valuable resource for researchers and clinicians interested in utilizing PBM therapy for arthritis treatment, providing a comprehensive overview of the existing literature and guiding future research in this field.

## 7. Conclusions and Prospects

In conclusion, this comprehensive review emphasizes PBM therapy’s potential as an effective and non-invasive arthritis treatment. The detailed overview covers the fundamental mechanisms of PBM treatment, elucidating its cellular function regulation and efficacy in small animal arthritis models. Although there is currently no universally agreed upon optimal PBM treatment parameters, the positive results of clinical trials and extensive research on the underlying mechanisms inspire confidence in the potential of this therapeutic approach. Further research should focus on determining the best parameters for PBM treatment of arthritis. This may involve investigating the effects of different wavelength ranges, dosages, and treatment durations on the treatment’s efficacy. Additionally, future clinical trials should rigorously employ standardized evaluation methods to ensure the safety and efficacy of PBM treatment.

In addition to establishing therapeutic parameters, the exact pathways by which PBM regulates arthritis treatment are not yet fully understood. Future studies could investigate these mechanisms further, particularly in relation to gene regulation and intercellular communication. Investigating these aspects can provide valuable insights that guide the selection of experimental parameters.

Furthermore, accurately assessing treatment efficacy is crucial for future research. Researchers can use multiple assessment methods, combining objective evaluations and subjective questionnaire surveys to enhance the reliability of treatment outcome conclusions, especially in clinical studies.

In summary, PBM therapy shows great promise as an effective and non-invasive treatment for arthritis. This review provides valuable insights and guidance for researchers interested in exploring the use of PBM therapy for arthritis treatment. With continued research and development, PBM therapy has the potential to become a widely adopted and beneficial treatment option for arthritis patients.

## Figures and Tables

**Figure 1 ijms-24-14293-f001:**
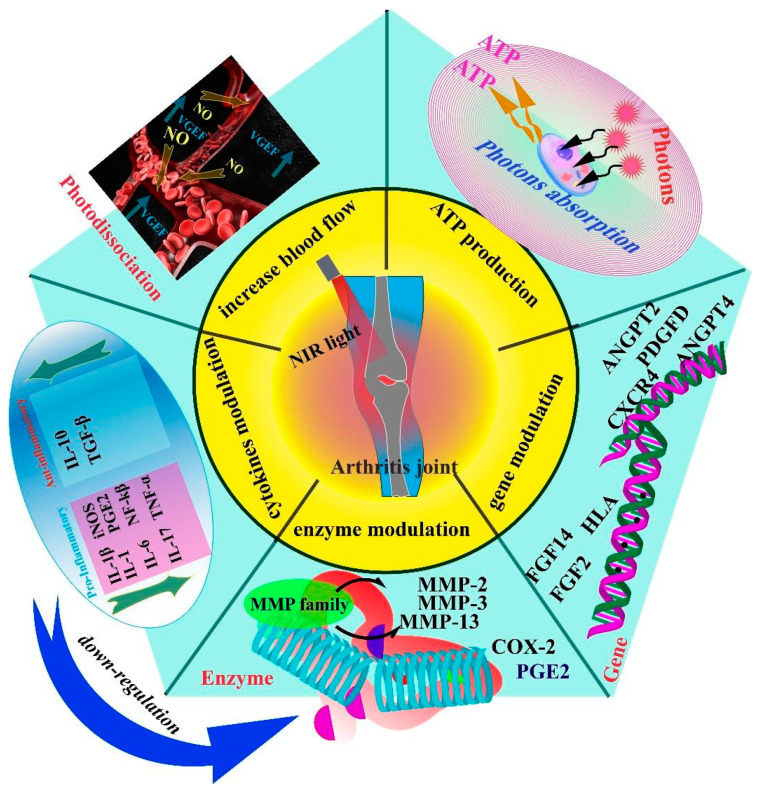
PBM exerts its therapeutic effects on arthritis through five key mechanisms: regulation of angiogenesis, stimulation of ATP production in cells, modulation of arthritis-related genes, regulation of the secretion of joint-related enzymes, and modulation of the expression of cytokines, including both pro-inflammatory and anti-inflammatory factors. These mechanisms collectively contribute to the efficacy of PBM treatment for arthritis. The regulation of angiogenesis helps reduce the infiltration of inflammatory cells and promotes increased blood flow, aiding in the management of arthritis. Increased ATP production enhances the function and activity of cells involved in arthritis. Modulating gene expression can modify the production of enzymes and inflammatory factors, leading to a reduction in the production of enzymes that damage joint tissues. The overall result is a decrease in inflammation and improvement in arthritis symptoms.

**Figure 2 ijms-24-14293-f002:**
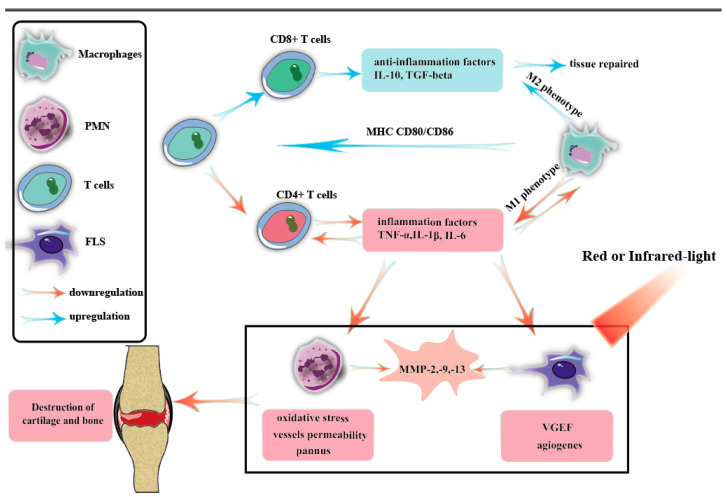
The activation of innate immunity is the underlying mechanism that triggers RA. Dendritic cells (DCs), macrophages, and activated B and T cells express major histocompatibility complex, CD80/CD86, and other inflammatory stimulatory proteins, which contribute to the differentiation of T cells into T helper 1 (Th1) and Th17 cell phenotypes. Once activated, T and B cells secrete inflammatory cytokines and chemokines, further activating leukocytes, macrophages, fibroblasts, and endothelial cells. This complex network of molecular and cytokine-mediated interactions forms the basis of RA pathogenesis. Each cell involved in the pathogenesis of RA plays a significant role in mediating inflammation and joint destruction. Therefore, inhibiting the function and response of these cells has become a crucial objective in alleviating or potentially curing RA.

**Table 1 ijms-24-14293-t001:** The mechanisms of the methods to arthritis induction.

Type of Arthritis	Methods	Mechanism
OA	surgical intervention	surgical methods, such as destabilization of the medial meniscus, can be used to create an animal model that mimics the mechanical stress and inflammation that occur in human OA.
	papain solution	papain is a proteolytic enzyme that can break down cartilage and induce arthritis in animal models. It is commonly used to create OA models.
	injecting small amounts of senescent cells	Injecting small amounts of senescent cells: senescent cells are cells that have stopped dividing but remain active and can secrete proinflammatory cytokines. Injecting small amounts of these cells can induce arthritis in animal models.
RA	CIA	collagen-induced arthritis (CIA) is an autoimmune disease model that involves immunization with type II collagen and adjuvant to induce an autoimmune response against the animal’s own cartilage.
	CFA	complete Freund’s adjuvant (CFA) is a non-specific immunostimulant that can induce inflammation and arthritis in animal models.
other types of arthritis	microcrystalline arthritis	microcrystalline compounds, such as monosodium urate and calcium pyrophosphate dihydrate, can induce arthritis in animal models by stimulating the immune system and causing inflammation.
	zymosan	zymosan is a polysaccharide derived from yeast cell walls that can activate the immune system and induce inflammation and arthritis in animal models.

**Table 2 ijms-24-14293-t002:** Study on PBM treatment of animal arthritis models.

Modeling Method	Wavelength (nm)	Power Density (mW/cm^2^)	Dose (J/cm^2^)	Light Source	Effect	Reference
The OA model was induced by anterior cruciate ligament transection	808	1700	50	(GaAlAs) diode laser	These results suggest that exercise training and LLLT were effective in preventing cartilage degeneration and modulating inflammatory processes induced by knee OA.	[77]
intradermal injection of an emulsion of bovine type II collagen and complete Freund’s adjuvant	570, 940	141.54	5	LED	the 940 nm light can inhibit the swelling of arthritis and reduce the production of IL-1β, IL-6 and MMP-3.	[133]
injection of zymosan	628, 685, 830	25	2.5	Laser (685 nm and 830 nm) and LED (628 nm)	Laser light significantly inhibited edema, vascular permeability and pain perception, while LED light had no effect.	[134]
injection of zymosan	830	200	3, 30	AlGaAs laser	PMN cells pro-apoptotic gene expression increased PBM at 30 J/cm^2^.	[89]
anterior cruciate ligament transection	630, 870	N/A	2 (red), 2.5 (IR)	LED	PBM effectively reduced inflammation levels in arthritis.	[78]
injection of CFA	880	4976.11	2985.668	laser	PBM improved cartilage defects and reduced inflammatory cells.	[135]
injection of collagenase	660, 780	750	7.5	laser	PBM can accelerate the initial breakdown of collagenase damaged cartilage and stimulate fibroblast synthesis of repaired collagen iii.	[136]
Papain-induced inflammation	808	1785.71, 3571.42	142.4	laser	50 mW LLLT is more effective in regulating inflammatory mediators (IL-1B, IL-6) and inflammatory cells (macrophages and neutrophils).	[76]
injection of CFA	658	69.11	N/A	diode laser	PBM reduced macrophage number and relieves swelling.	[137]
injection of papain solution	632	3.1	2.79	He-Ne lasers	PBM enhanced the biosynthesis of arthritic cartilage and lead to the improvement of arthritic histopathological changes.	[121]
bone defect by motorized round drill	830	2653.92	249.47	laser	PBM may promote the development of new bone by regulating the expression of inflammatory genes and angiogenic genes as well as the immune expression of COX2 and VEGF in the early stage of bone healing.	[67]
injection of papain solution	660, 808	3570	142.8	InGaAlP (660 nm), AsGaAl (808 nm)	PBM on 808 nm laser promoteed angiogenesis and reduces the formation of fibrosis.	[122]
injection of CFA	780	150 and 400	4.5 and 72	GaAlAs laser	PBM effectively reduced the inflammatory response and FLS cell number in the inflammatory site.	[110]
surgical procedure	780	500	10	AsGaAl diode laser	PBM decreased the release of TNF-α and IFN-γ in monocytes.	[71]
Induced microcrystalline arthritis	670	1500	18	InGaAlP laser	PBM reduced the synovium inflammatory process and tissue injuries.	[82]
Bilateral translumbar aortic incision	830	3530 and 7060	60 and 12	GaAlAs laser	PBM stimulated new bone formation, vascular fibrosis, and angiogenesis.	[126]
collagenase injection	660	3570	35.71 and 107.14	infrared laser unit	PBM reduced the production of pro-inflammatory markers such as IL-6 and TNF-α.	[72]
collagen-induced arthritis	830	6.4	7.64	Ga-Al-As diode laser device	PBM inhibited the CCL2 productions in CIA rats.	[138]
injection of zymosan	810	5 and 50	3 and 30	A diode laser	PBM reduces PGE2 production, and long-term PBM is more effective in treating arthritis.	[139]
collagen-induced arthritis	780 nm	100	7.7	laser	PBM could modulate the inflammatory response both in early as well as in late progression stages of RA.	[140]
injection of CFA	830	250	3	GaAlAs laser	PBM could increase the remodeling and enhancing tissue repair in arthritis.	[141]
injection of zymosan	830	200	30	(GaAlAs) low-level infrared laser	PBM could alter the inflammatory state and stimulate immune cells to accelerate the relief of arthritis symptoms.	[70]
surgical procedure	660 and 780	1750	25.025	AlGaInP (660 nm) and AlGaAs (780 nm)	PBM was able to modulate the inflammation phase, optimize the transition from the inflammatory to the regeneration phase and improve the final step of regeneration, enhancing tissue repair.	[98]
Inducing microcrystalline arthritis	632.8	200	8	He-Ne Laser	PBM reduced TNF-α and PGE2 production in the treatment of microcrystalline arthritis.	[125]
injection of CFA	660 nm	500	5	InGaAIP	PBM could reduce leukocyte migration and restore joint function.	[142]
injection of zymosan	660	250	2.5	InGaAIP	PBM could effectively reduce inflammation and inhibit collagen degradation.	[93]
collagen-induced arthritis	830	100	5	Ga-Al-As diode laser device	PBM effectively reduced CXCR4 gene expression.	[91]

Abbreviation: IR (Infrared Red), N/A (Not Applicable).

**Table 3 ijms-24-14293-t003:** The arthritis evaluation methods in clinic trials.

Class	Evaluate Methods	Performance
subjective judgment	VAS	The pain score ranges from 1 to 10, with 0 indicating no pain and 10 indicating the most severe pain
	DASH	DASH is a patient assessment of upper limb function. The two-part rating scale contains 30 indicators, mainly by examining activities related to daily life, including the degree of limitation of living ability and social activity ability
	HAQ	The content of the survey is to select and score each joint site according to its difficulty in daily life.
	WOMAC	The structure and function of the hip and knee were evaluated from pain, stiffness and joint function, and the functional description was mainly aimed at the lower limbs.
objective judgment	ROM	A universal goniometer will be used to measure the angular range of knee motion in patients with knee arthritis
	Peripheral blood assay	The contents of IL-6, NK-KB, C-reactive protein and erythrocyte sedimentation rate in peripheral blood were determined.
	Radiographic analysi	The structure of the patient’s joints was reconstructed by X-CT and MRI and so on.
	stiffness time	The longer the morning stiffness, the worse the arthritis

**Table 4 ijms-24-14293-t004:** The studies of PBM on the clinic arthritis trials.

Types of Arthritis	Objects	Wavelengths (nm)	Power Density (mW/cm^2^)	Evaluation Methods	Effects	Reference
RA	82 RA patients	785	70	VAS, HAQ and DASH	Low levels of aluminum-gallium arsenide laser therapy were ineffective in wavelength, dose, and power for hand treatment in patients with rheumatoid arthritis.	[22]
	60 RA patients	808	100	HAQ, ELISA, and blood routine test	It could be concluded that laser acupuncture is effective in the adjuvant treatment of elderly rheumatoid arthritis.	[79]
	18 RA patients	633	10	ROM, VAS and morning stiffness (MS)	Of these cases, 2 had a positive development on the joint ability score but not on the VAS or MS, while the third patient had an improvement on the VAS but not on the other parameters.	[170]
	25 female RA patients	820	50	VAS, Blood test for hand function	A total of 72 percent of patients reported pain relief. There were no significant changes in other clinical, functional, scintillation, or laboratory features.	[169]
OA	One hundred consecutive randomly selected elderly patients with bilateral symptomatic knee arthritis	810	20	WOMAC	After 6 years of follow-up, patients in the PBM group benefited significantly, only 1 patient needed joint replacement, and 9 patients in the non-PBM group needed surgery (*p* < 0.05).	[151]
	22 female and 5 male knee OA patients	830	50	VAS, blood and urine tests	The results showed that PBM could alleviate pain and improve microcirculation in the irradiated area of KOA.	[159]
	145 patients aged 50 to 75 years with knee joint	904	40	WOMAC, VAS, ROM	The specific PBM parameters need to be better defined.	[172]
	40 knee OA patients	850	50	WOMAC, and VAS	PBM appeared to be an effective method for short-term pain relief and functional improvement in patients with chronic knee osteoarthritis.	[160]
	90 knee OA patients	904	10–20	WOMAC, and VAS	The use of different doses and durations of PBM did not affect the results, and both regimens are safe and effective for the treatment of knee OA.	[161]
	40 knee OA patients	904	60	WOMAC, VAS, muscle strength testing and ROM	PBM combined with exercise can effectively relieve pain, function, and mobility in patients with knee osteoarthritis.	[162]
	49 knee OA patients	830	30	VAS	PBM applied to specific acupoints in a short period of time, together with exercise, can effectively reduce the pain of knee osteoarthritis patients and improve their quality of life.	[163]
	53 male knee OA patients	830	50	WOMAC and VAS	In the treatment of patients with osteoarthritis, high-intensity laser therapy combined with exercise was more effective than low-intensity laser therapy combined with exercise, and both treatments were better than exercise alone.	[164]
	60 knee OA patients	830	50	WOMAC	PBM had no effect on pain in patients with knee OA.	[20]
	34 knee OA patients (32 female)	658, 785	40	VAS, and ROM	PBM is a safe, noninvasive, efficient, and effective method to reduce pain, swelling, and increase joint mobility in patients with Heboden and Buchar osteoarthritis.	[165]
	34 chronic knee OA patients	905	500	VAS, X-CT, Biochemical test analysis (urine)	PBM treatment is effective in reducing pain and improving cartilage thickness through biochemical changes.	[158]
	88 hands OA patients	860	3000	VAS, and ROM	PBM was no better than placebo in reducing pain, morning stiffness, or improving functional status.	[21]
	37 hands OA patients	808	40	Pain stress tests, hand grip tests	PBM did not improve hand grip strength significantly, but it reduces the pain.	[171]
	47 knee OA patients	904	60	VAS, HAQ, and ROM	PBM can improve pain and function in patients with knee osteoarthritis in the short term.	[166]
	36 female and 14 males had degenerative osteoarthritis in both knees	633, 830	8–122	VAS	PBM is effective in relieving pain and disability in degenerative knee osteoarthritis.	[171]
Other	61 patients with lower back pain for at least 12 weeks	810	50	VAS and ROM	In chronic low back pain, PBM combined with exercise is more beneficial than long-term exercise alone.	[168]

## Data Availability

No new data were created or analyzed in this study. Data sharing is not applicable to this article.

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
