# Peer review of "The Mechanisms and Efficacy of Photobiomodulation Therapy for Arthritis: A Comprehensive Review"

_ijms, 2023, doi:10.3390/ijms241814293_

Round 1

Reviewer 1 Report

Dear Authors,

I have read with interest the submitted manuscript.

I believe that the chosen topic covers and interesting filed in the management of arthritis.

I would consider the publishing the manuscript in its current form.

Author Response

Dear Reviewer,

Thank you for your positive feedback and support of our review article. Your recognition is important to us and will motivate us to enhance the manuscript's quality and credibility in the scientific community. We are extremely grateful for your valuable comments and suggestions, which will help us refine the article further.

Reviewer 2 Report

Dear Editor, I have reviewed the manuscript 'The Mechanisms and Efficacy of Photobiomodulation Therapy 2 for Arthritis: A Comprehensive Review' submitted for IJMS journal. The paper is a comprehensive review regarding the use and outcomes of photobiomodulation using near-infrared light in treating pain and disability associated to osteoarthritis. The subject is of novelty, and, while there are solid evidences that photomodulation is an efficient therapy in osteoarthritis, there are still challenges regarding the optimal parameters. The manuacript is based on an impressive number of referrences (194) that comprises all relevant researches on this subject The effect of photomodulation is discussed from the cellular and molecular point of view, the authors synthetise correctly what are the mechanism of action that make this therapy efficient. The tables and figures are well explained As minor issues: 1. A paragraph should be added in the introduction what were the databases used for this research and the time frame for the selected studies. 2. A paragraph with current challenges and future directions could be added at the Discussion.

Author Response

Dear Reviewer, thank you for acknowledging our review article and providing valuable suggestions. Your input has greatly contributed to improving this review. We have now made the following modifications based on your recommendations:

  1. Method: The search terms "low-level laser therapy," "rheumatoid arthritis disease," and "photobiomodulation" were entered into Google Scholar, PubMed, and Medline. The search for PBM studies on arthritis spanned from 1987 to 2022. These statements have been added at line 73 in the introduction.

  1. With regards to the challenges and future directions outlined in the manuscript concerning PBM in arthritis treatment, we have made the subsequent revisions to address the reviewer's comments:

In line 747, we have included the following sentence: "Although there is currently no universally agreed upon optimal PBM treatment parameters, the positive results of clinical trials and extensive research on the underlying mechanisms inspire confidence in the potential of this therapeutic approach" to highlight the ongoing debate and divergence in PBM parameter settings.

We have made the suggested change to the sentence in line 754: "In addition to establishing therapeutic parameters, the exact pathways by which PBM regulates arthritis treatment are not yet fully understood. Future studies could investigate these mechanisms further, particularly in relation to gene regulation and intercellular communication. Investigating these aspects can provide valuable insights that guide the selection of experimental parameters." to highlight the significance of researching the mechanisms through which PBM modulates arthritis treatment.

Additionally, we emphasize the importance of precise treatment assessment, and in line 759, we state that "Accurately assessing treatment efficacy is crucial for future research. Researchers can use multiple assessment methods, combining objective evaluations and subjective questionnaire surveys to enhance the reliability of treatment outcome conclusions, especially in clinical studies."

With these adjustments, the review clearly identifies the present obstacles in studying the dose-effect correlation of PBM and investigating its particular mechanisms, whilst also underscoring the significance of appropriate evaluation techniques.

The attachment is the revised PDF version of the manuscript, and the changes were highlighted.

Reviewer 3 Report

The review is well written and have covered all  topics under each section. 

Few typographical and grammatical errors do exist. The authors should rectify this.

Author Response

 Respond:

We would like to express our gratitude to your meticulous evaluation of our review article. Your keen attention to detail has helped us identify and address these minor issues, enhancing the overall quality of our paper. Thank you for your valuable input and constructive feedback.

  1. In line 625, ‘rely on the patient’s subjective judgment through questionnaire responses to draw conclusion’ was deleted.

2 The spelling and grammar errors in Table 2 have been corrected. Details are highlighted in the text.

3 The spelling and grammar errors in Table 4 have been corrected. Details are highlighted in the text.

The attachment is the revised PDF of the manuscript, and the changes were highlighted.
